# Host immunity increases *Mycobacterium tuberculosis* reliance on cytochrome *bd* oxidase

Yi Cai[1◉], Eleni Jaecklein[2◉], Jared S. Mackenzie[3], Kadamba Papavinasasundaram[2], Andrew J. Olive[4], Xinchun Chen[1], Adrie J. C. Steyn[3], Christopher M. Sassetti[2]*

**1** Guangdong Provincial Key Laboratory of Regional Immunity and Diseases, Department of Pathogen Biology, Shenzhen University School of Medicine, Shenzhen, China, **2** Department of Microbiology and Physiological Systems, University of Massachusetts Medical School, Worcester, Massachusetts, United States of America, **3** Africa Health Research Institute, Durban, South Africa, **4** Department of Microbiology & Molecular Genetics, Michigan State University, East Lansing, Michigan, United States of America

◉ These authors contributed equally to this work.
\* Christopher.sassetti@umassmed.edu

**Data Availability Statement:** All relevant data are within the manuscript.

**Funding:** This work was supported by the National Institutes of Health (grant AI32130 to C.M.S.),

## Abstract

In order to sustain a persistent infection, *Mycobacterium tuberculosis* (*Mtb*) must adapt to a changing environment that is shaped by the developing immune response. This necessity to adapt is evident in the flexibility of many aspects of *Mtb* metabolism, including a respiratory chain that consists of two distinct terminal cytochrome oxidase complexes. Under the conditions tested thus far, the $bc_1/aa_3$ complex appears to play a dominant role, while the alternative *bd* oxidase is largely redundant. However, the presence of two terminal oxidases in this obligate pathogen implies that respiratory requirements might change during infection. We report that the cytochrome *bd* oxidase is specifically required for resisting the adaptive immune response. While the bd oxidase was dispensable for growth in resting macrophages and the establishment of infection in mice, this complex was necessary for optimal fitness after the initiation of adaptive immunity. This requirement was dependent on lymphocyte-derived interferon gamma (IFNγ), but did not involve nitrogen and oxygen radicals that are known to inhibit respiration in other contexts. Instead, we found that Δ*cydA* mutants were hypersusceptible to the low pH encountered in IFNγ-activated macrophages. Unlike wild type *Mtb*, cytochrome *bd*-deficient bacteria were unable to sustain a maximal oxygen consumption rate (OCR) at low pH, indicating that the remaining cytochrome $bc_1/aa_3$ complex is preferentially inhibited under acidic conditions. Consistent with this model, the potency of the cytochrome $bc_1/aa_3$ inhibitor, Q203, is dramatically enhanced at low pH. This work identifies a critical interaction between host immunity and pathogen respiration that influences both the progression of the infection and the efficacy of potential new TB drugs.

## Author summary

Tuberculosis, caused by *Mycobacterium tuberculosis* (*Mtb*), is a serious global health problem that is responsible for over one million deaths annually, more than any other single

National Natural Science Foundation of China (grant 82072252 to Y.C.), and the Arnold and Mabel O. Beckman Foundation (A.J.O.). The funders had no role in study design, data collection and analysis, decision to publish, or preparation of the manuscript. NIH: https://www.nih.gov/ Beckman Foundation: https://www.beckman-foundation.org/.

**Competing interests:** The authors have declared that no competing interests exist.

infectious agent. In the host, *Mtb* can adapt to a wide variety of immunological and environmental pressures which is integral to its success as a pathogen. Accordingly, the respiratory capacity of *Mtb* is flexible. The electron transport chain of *Mtb* has two terminal oxidases, the cytochrome $bc_1/aa_3$ super complex and cytochrome $bd$, that contribute to the proton motive force and subsequent production of energy in the form of ATP. The $bc_1/aa_3$ super complex is required for optimal growth during infection but the role of cytochrome $bd$ is unclear. Here we report that the cytochrome $bd$ oxidase is required for resisting the adaptive immune response, in particular, acidification of the phagosome induced by lymphocyte-derived IFNγ. We found that the cytochrome $bd$ oxidase is specifically required under acidic conditions, where the $bc_1/aa_3$ complex is preferentially inhibited. Additionally, we show that acidic conditions increased the potency of Q203, a cytochrome $bc_1/aa_3$ inhibitor and candidate tuberculosis therapy. This work defines a new link between the host immune response and the respiratory requirements of *Mtb* that affects the potency of a potential new therapeutic.

## Introduction

Tuberculosis (TB) is responsible for an estimated 1.4 million deaths annually and remains one of the most deadly infectious diseases [1]. The causative agent of TB, *Mycobacterium tuberculosis* (*Mtb*), is an obligate aerobe and relies on oxidative phosphorylation (OXPHOS) via the electron transport chain (ETC) and glycolysis for energy production. The mycobacterial ETC has two terminal oxidases, the cytochrome $bc_1/aa_3$ super complex that is related to mitochondrial complex III and IV, and the cytochrome $bd$ oxidase which is unique to prokaryotes. These terminal oxidases transfer electrons from the ETC to $O_2$ and contribute to the proton motive force (PMF) gradient that powers the production of ATP by ATP synthase. Both genetic [2] and chemical inhibition of the cytochrome $bc_1/aa_3$ [3–5] has been used to show that this complex is required for optimal growth and persistence during infection, and cytochrome $bc_1/aa_3$ inhibitors are under evaluation as antimycobacterial therapies [6].

In the absence of cytochrome $bc_1/aa_3$, electrons are rerouted through the cytochrome $bd$ oxidase [7]. The latter complex in *Mtb* is encoded in a single operon, *cydABDC*, which produces both the *cydAB* oxidase complex and *cydDC*, a putative ABC-transporter that has not been studied in *Mtb*, but is necessary for assembly of the cytochrome in *Escherichia coli* [8,9]. Genetic deletion of the *cydABDC* operon produces hyper-susceptibility to cytochrome $bc_1/aa_3$ inhibitors, demonstrating a partially-redundant role for the terminal oxidases [4,7,10]. However, the specific role played by the cytochrome $bd$ oxidase in *Mtb* remains unclear. In *E.coli*, the cytochrome $bd$ oxidase detoxifies peroxide radicals and maintains respiration under hypoxic conditions [11,12]. Similar observations in the saprophyte, *Mycobacterium smegmatis*, show that *cyd* mutants are hyper-susceptible to peroxide stress and expression of the *cydAB* operon is induced in hypoxic conditions [13–15]. In addition, a transposon mutant screen predicted that the *cydABDC* operon is required for optimal *Mtb* growth at pH 4.5, suggesting additional functions [16]. While it is plausible that these properties contribute to *Mtb* fitness during infection, the role played by the cytochrome $bd$ oxidase in the mouse model of TB remains unclear. Some studies report no effect of *cydABDC* mutation, while others describe a fitness defect at the later stages of infection [2,4,17]. Thus, while it is clear that the cytochrome $bd$ oxidase is active in mycobacteria, the non-redundant role of this system during infection is unknown.

As an obligate aerobe it is likely that *Mtb*'s flexible respiratory chain has evolved to adapt to the changing environment encountered during infection. During the initial days after infection of the lung, *Mtb* replicates in macrophages, but once these cells are stimulated by T cell-derived cytokines, they restrict *Mtb* growth. The stressors associated with activation of the macrophage cause a number of specific alterations in the bacterial environment that may alter respiratory requirements. In particular, IFNγ induces antimicrobial responses in the macrophage, including the production of the known respiratory poison, nitric oxide (NO) [18] via nitric oxidase synthase 2 (NOS2). Additionally, IFNγ induces superoxide production via the NADPH-dependent phagocyte oxidase (Phox), which alters the respiratory requirements of another intracellular pathogen, *Salmonella enterica* [19]. Lastly, IFNγ promotes the maturation of the pathogen-containing vacuole, promoting both its acidification and fusion with more degradative compartments. The observation that *cydAB* expression peaks with the onset of the adaptive immune response in the mouse model of infection further suggests an association between T cell cytokines, such as IFNγ, and alterations in the respiratory requirements of *Mtb* [17].

In this work, we investigated the interactions between macrophage activation and the mycobacterial respiratory chain. We report that cytochrome *bd* oxidase is specifically required for the bacillus to resist IFNγ-induced macrophage function. In particular, cytochrome *bd* oxidase is necessary in acidic environments similar to those encountered in the phagosome of IFNγ activated macrophages. These compartments can reach pH levels as low as 4.5 [20,21], which we show preferentially inhibits the function of the cytochrome $bc_1/aa_3$ complex. The relative acid-resistance of the cytochrome *bd* oxidase explains its role in counteracting IFNγ-dependent immunity, and suggests important interactions between immunity and respiratory chain inhibitors that are in clinical development as TB therapeutics.

## Results

### Δ*cydA* mutant is susceptible to IFNγ-activation of macrophages independent of NOS2 and Phox

To investigate the effects of macrophage activation state on the requirement for the cytochrome *bd* oxidase in *Mtb*, we constructed a Δ*cydA* deletion mutant in H37Rv [22]. Consistent with previous studies, there was no difference in growth between H37Rv and Δ*cydA* mutants in broth culture [2,10] (Fig 1A). We compared the fitness of H37Rv and Δ*cydA* mutants in bone marrow-derived macrophages (BMDMs) from C57BL/6J (wildtype) mice. Initially, we used flow cytometry and fluorescent live/dead reporter strains of *Mtb* to estimate relative bacterial growth and viability. The *Mtb* strains expressed a constitutive GFP marker and an anhydrotetracycline (ATc)-inducible RFP marker. GFP intensity was used to estimate total infected cell number and ATc-induced RFP intensity served as a surrogate measure of the relative viability of the bacterial population in each infected macrophage, and has been show to correlate with CFU in this setting [23]. By these metrics, the growth and viability of H37Rv and the Δ*cydA* mutant were not appreciably different in unstimulated BMDMs (Fig 1B).

In other bacterial systems, the cytochrome *bd* oxidase is important for resistance to NO and oxidative stress, which are major mediators of IFNγ-dependent antimicrobial activity [11,24–26]. To determine if IFNγ or these reactive species alter the requirement for Δ*cydA*, we stimulated BMDMs with IFNγ, and included cells from $Nos2^{-/-}$ and $Cybb^{-/-}$ mice which lack functional NOS2 and Phox systems, respectively. In wildtype BMDMs, addition of IFNγ significantly reduced the number of cells harboring live H37Rv and Δ*cydA* bacteria (Fig 1B). IFNγ treatment had no effect on the viability of H37Rv in $Nos2^{-/-}$ BMDMs, indicating that IFNγ-mediated inhibition of H37Rv is primarily dependent on NO, consistent with previous studies

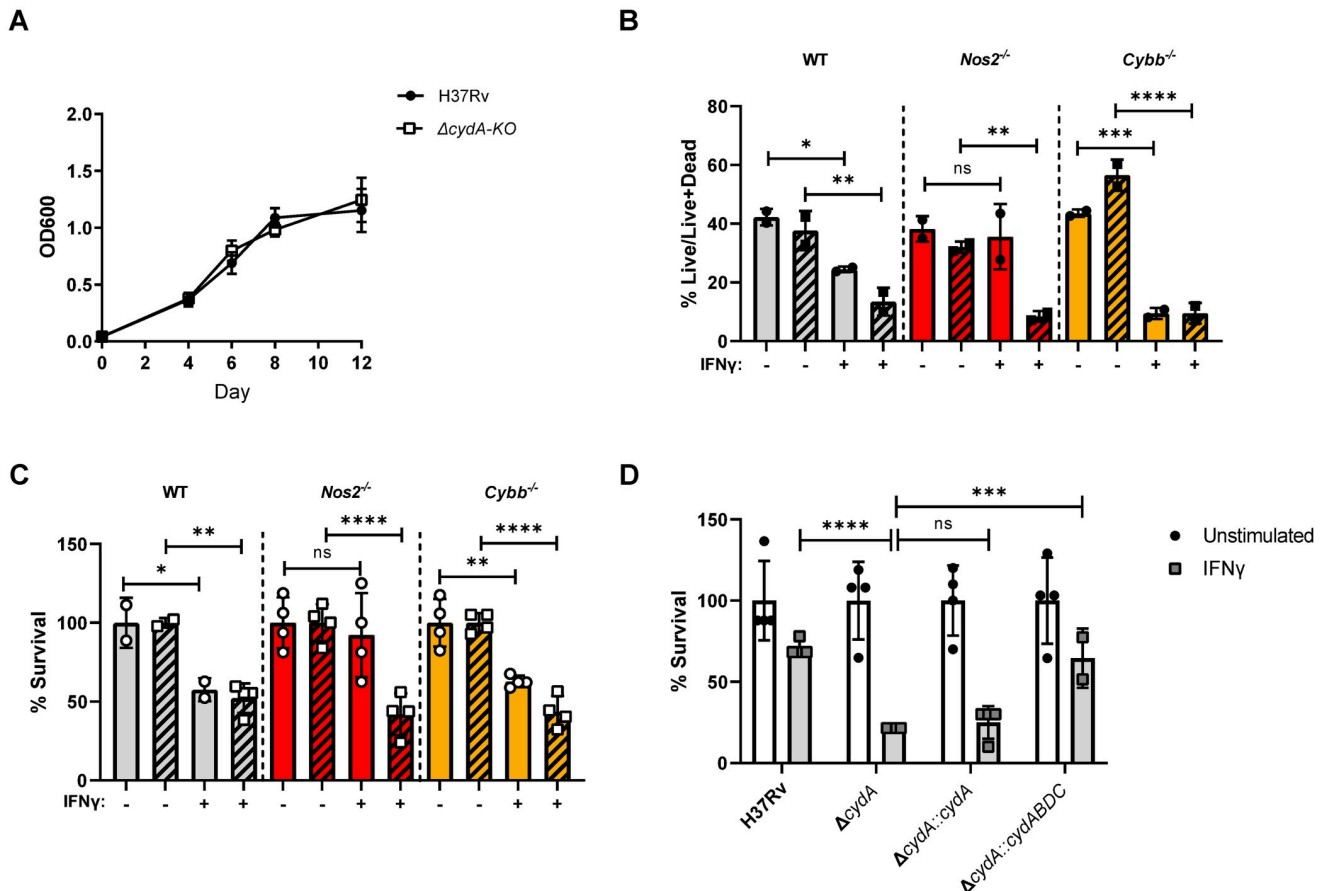

**Fig 1. CydA is required to resist IFNγ-mediate immunity independent of NOS2 and Phox.** A) Growth of H37Rv and *ΔcydA* in 7H9 broth over 12 days in 96 well plates. B) C57BL/6J (WT), *Nos2*^−/−^ and *Cybb*^−/−^ BMDMs were left untreated or treated with 25ng/mL of IFNγ for 18 h. Macrophages were infected with H37Rv (solid) or *ΔcydA* (diagonal bars) live-dead reporter strains (MOI = 5). Y-axis represents the fraction BMDMs with live bacteria (RFP⁺GFP⁺) over total infected BMDMs (GFP⁺) determined by flow cytometry at 4 days post-infection. C) IFNγ treated or untreated BMDMs were infected (MOI = 5) with H37Rv (solid) or *ΔcydA* (diagonal bars). CFU determined 4 days post-infection. Data were normalized using the following formula, [sample value]/[mean of untreated]*100, error bars reflect the normalized values. Mean of untreated for H37Rv in WT, *Nos2*^−/−^, and *Cybb*^−/−^ BMDMs: 9.9x10⁴, 1.28x10⁵, and 8.5x10⁴ CFU/mL, respectively. Mean of untreated for *ΔcydA* in WT, *Nos2*^−/−^, and *Cybb*^−/−^ BMDMs: 4.7x10⁴, 1.0x10⁵, and 7.45x10⁴ CFU/mL, respectively. D) IFNγ treated or untreated *Nos2*^−/−^ BMDMs infected with the indicated strains (MOI = 5). CFU determined 4 days post-infection. Represented as percent survival relative to untreated, as in panel C. Mean of untreated for H37Rv 4.1x10⁴ CFU/mL, *ΔcydA* 3.7 x10⁴ CFU/mL, *ΔcydA::cydA* 4.0 x10⁴ CFU/mL, and *ΔcydA::cydABDC* 3.1 x10⁴ CFU/mL. Analysis of B-D was preformed using one-way ANOVA with Sidak post-test to correct for multiple comparisons. * p-value < 0.05, ** p-value < 0.01, *** p-value < 0.001, **** p-value <0.0001. Data depict single experiments that are representative of at least 2 independent studies.

[21,25] (Fig 1B). In contrast, NOS2 and CYBB were not necessary for IFNγ to inhibit *ΔcydA* mutants (Fig 1B).

These observations were confirmed by CFU enumeration (Fig 1C). The magnitude of IFNγ-dependent inhibition was smaller in the CFU assay than the flow cytometry study, likely reflecting increased sensitivity of the live/dead reporter for bacterial fitness. Regardless, the CFU assay also showed that IFNγ treatment reduced the viability of H37Rv in a *Nos2*-dependent, *Cybb*-independent manner, whereas the suppression of *ΔcydA* mutants was independent of both mediators. This *ΔcydA* mutant phenotype could be complemented *in trans* by expressing the *cydABDC* operon but not *ΔcydA* alone (Fig 1D). These observations indicated the entire *cydABDC* operon was necessary to resist an IFNγ-induced stress that is independent of NOS2 and CYBB.

## IFNγ but not iNOS or Phox is necessary for the attenuation of *ΔcydA* in mouse lungs

The interaction between IFNγ and *ΔcydA* was then assessed in the mouse model. To evaluate the relative fitness of the *ΔcydA* mutant, we performed competitive infections using a mixture of H37Rv::Kan and *ΔcydA*::Hyg. To test the importance of lymphocyte-derived IFNγ, the relative fitness of the *ΔcydA* mutant was assessed in wild type C57BL/6J mice and animals lacking adaptive immunity (*Rag2*$^{-/-}$), IFNγ receptor (*Ifngr1*$^{-/-}$), NOS2 (*Nos2*$^{-/-}$) and Phox (*Cybb*$^{-/-}$) (Fig 2A). At each timepoint, lung homogenates were plated on kanamycin and hygromycin and CFU were enumerated to compare the fitness of H37Rv and *ΔcydA* (Fig 2B). Day 1 CFU showed equivalent numbers of H37Rv and *ΔcydA* bacteria in the lung (Fig 2C). On day 15 post infection, before the onset of adaptive immunity, there was no significant difference in CFU between H37Rv and *ΔcydA* across all five mouse genotypes (Fig 2B). Once the adaptive response was established after 30 days of infection, we observed a significant decrease in *ΔcydA* lung CFU compared to H37Rv in wildtype, *Nos2*$^{-/-}$, and *Cybb*$^{-/-}$ mice. However, there was no difference between *ΔcydA* and H37Rv lung CFU in *Ifngr1*$^{-/-}$ and *Rag2*$^{-/-}$ mice demonstrating that the attenuation of the *ΔcydA strain* is dependent on lymphocytes and IFNγ (Fig 2B). Complementation of the *ΔcydA* mutant with *cydABDC* rescued the fitness defect observed in the mutants at the 30-day timepoint in wildtype mice (Fig 2C). These observations were consistent with the *ex vivo* macrophage infections, both indicating that the IFNγ-dependent attenuation of *ΔcydA* is NOS2 and Phox independent.

## *ΔcydA* mutants are defective for growth at low pH

Beyond stimulating the production of RNS and ROS in macrophages, IFNγ also promotes the maturation and acidification of the mycobacterial phagosome. An *in vitro* transposon mutant screen suggested that the *cydABDC* operon may contribute to optimal *Mtb* growth at pH 4.5 [16]. To determine if the hyper-susceptibility of *ΔcydA* to IFNγ could be due to phagosomal maturation and acidification, we investigated the fitness of *ΔcydA* mutants under acidic conditions. 7H9 media was adjusted to pH values that span those encountered in the maturing phagosome [20,21] using citrate-phosphate buffer, as described [27,28]. Using a multi-well plate assay, we found that the growth rate of *ΔcydA* was significantly reduced at low pH in comparison to H37Rv (Fig 3A and 3B). However, we also observed that the *ΔcydA* mutant grew modestly faster than H37Rv at pH 7.4 in this static assay format. To ensure equal aeration of these cultures, we also conducted growth studies in agitated cultures using with H37Rv, *ΔcydA* and the complemented strain. In this assay format, we again observed that the growth of *ΔcydA* was significantly attenuated at pH 6.2 compared to either the parental H37Rv strain or the complemented mutant (*ΔcydA*::*cydABDC)* (Fig 3C).

We next sought to determine if the IFNγ-dependent acidification of the phagosome in macrophages could account for the intracellular growth defect of the *ΔcydA* mutant. BMDMs from wildtype, *Nos2*$^{-/-}$ and *Cybb*$^{-/-}$ mice were infected with either H37Rv or the *ΔcydA* mutant, and we determined if the inhibitory effect of IFNγ was altered by bafilomycin A1, an inhibitor of the vacuolar-type H$^+$-ATPase that is responsible for the acidification and maturation of the phagosome. Using the live/dead reporter as a surrogate for bacterial fitness, we confirmed that NOS2 was required for IFNγ to inhibit H37Rv, but not the *ΔcydA* mutant, providing a situation where the NO-independent inhibitory effect of IFNγ on *bd* oxidase-deficient *Mtb* could be assessed. In these *Nos2*$^{-/-}$ macrophages, BAF had no effect on the fitness of H37Rv, but completely reversed the inhibitory effect of IFNγ on the *ΔcydA* mutant, restoring fitness to levels equivalent to unstimulated BMDMs (Fig 3D). In all three macrophage genotypes, BAF treatment restored the fitness of the *ΔcydA* mutant to wild type levels in the presence of IFNγ.

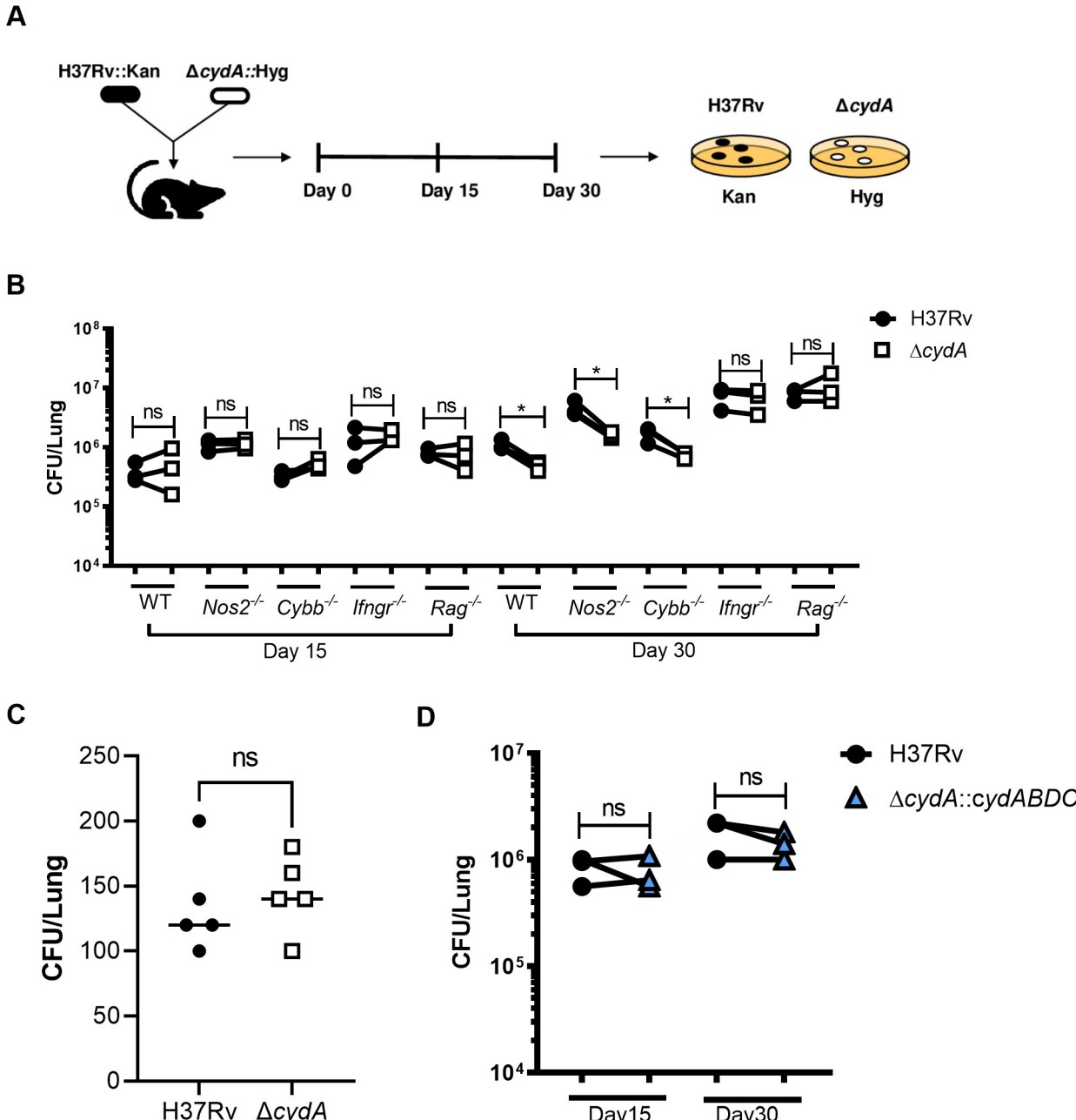

**Fig 2. CydA is required for persistence in IFNγ competent mice independent of Nos2 or Phox.** A) Schematic of experimental design for 1:1 co-infection with H37Rv::Kan and Δ*cydA*::Hyg. Lungs were collected on day 15 and day 30 post infection and dilutions of lung homogenates were plated on both 7H10+kanamycin and 7H10+hygromycin. B) CFU of H37Rv (black) and Δ*cydA* (open) in the lungs of C57BL/6J (WT), *Nos2*−/−, *Cybb*−/−, *Ifngr1*−/−, and *Rag2*−/− mice were enumerated at day 15 and day 30 post-infection. C) Day 1 H37Rv and Δ*cydA* CFU from C57BL/6J mice. D). Lung CFU of H37Rv and the complemented mutant strain (Δ*cydA*::*ABDC)* shown at day 15 and day 30 post-infection. All comparisons assessed using paired t-test. * p-value < 0.05. Data depict single experiment that is representative of at least 2 independent studies.

While BAF treatment had a modest effect on the fitness of wild type *Mtb*, the preferential effect on the Δ*cydA* mutant was consistent with the hypersensitivity of this strain to low pH. Together these observations indicate that the NO-independent inhibitory effect of IFNγ on *bd* oxidase-deficient *Mtb* could primarily be attributed to the environment encountered in the mature acidified phagosome.

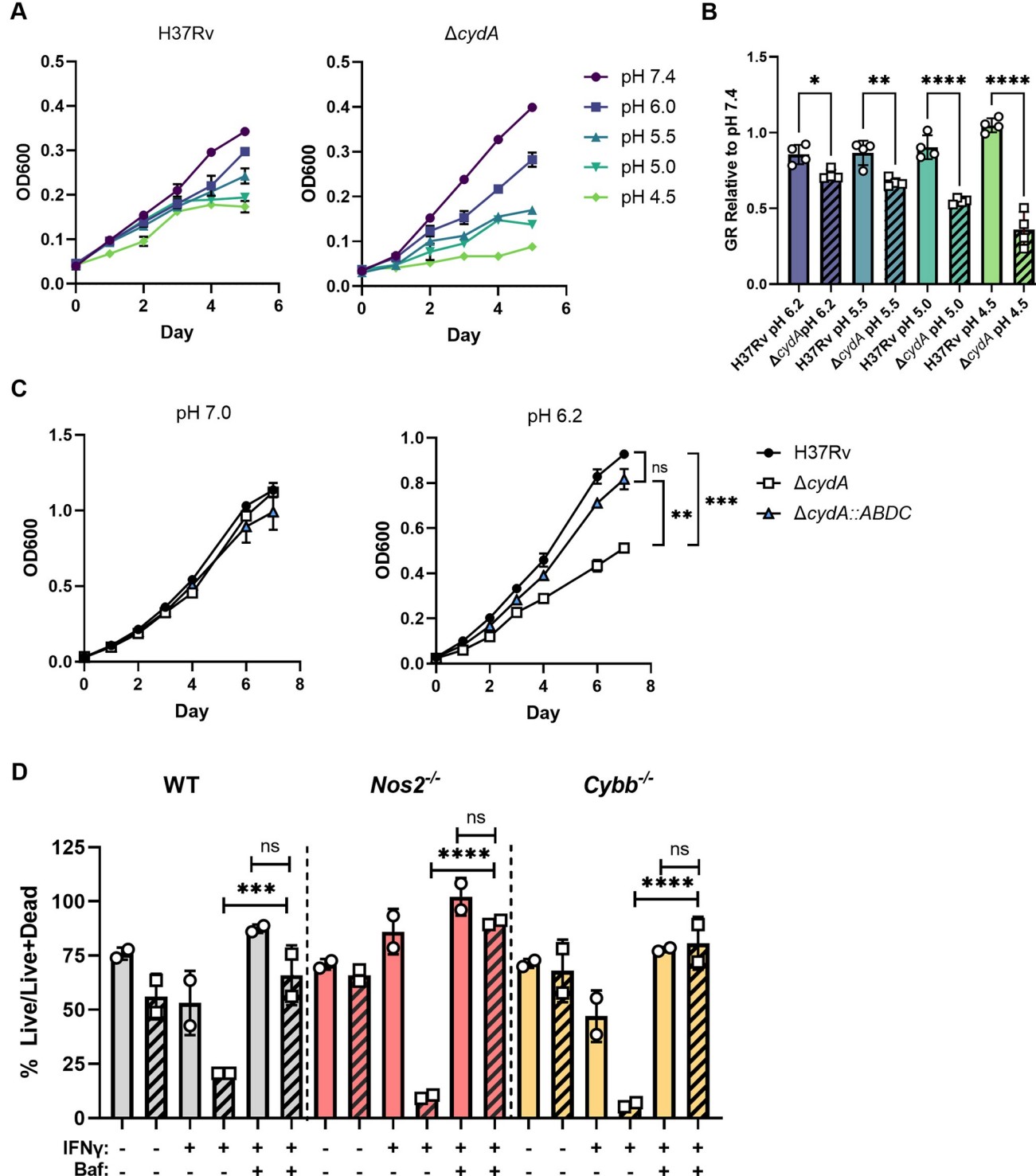

**Fig 3. CydA is required for growth in acidic conditions.** A) Growth of H37Rv and $\Delta cydA$ in 7H9-tyloxapol at pH 7.4, pH 6.0, pH 5.5, pH 5.0, and pH 4.5 measured by OD600 for 5 days in a 96 well plate. B) Growth rate (GR) for the samples in panel A, relative to pH 7.4. GR was calculated between days 0 and 4, when the increase in OD was linear. C) Growth of H37Rv, $\Delta cydA$, and $\Delta cydA::ABDC$ in 7H9-Tyloxapol at pH 7.0, pH 6.2, measured by OD600 for 6 days in aerated (i.e.agitated) cultures. D) WT, $Nos2^{-/-}$ and $Cybb^{-/-}$ BMDMs were left untreated or treated with IFNγ (25ng/mL). Macrophages were infected with H37Rv (solid) or $\Delta cydA$ (diagonal bars) live-dead reporter strains (MOI = 5). Post-infection BMDMs were left untreated, treated with IFNγ, or treated with IFNγ and Bafilomycin A (100ng/mL). The fraction of macrophages harboring live bacteria (%Live/ Live+Dead) was determined using flow cytometry. Analysis of B-D was performed using a one-way ANOVA with Sidak post-test to correct for multiple comparisons. * p-value < 0.05, ** p-value < 0.01,

[**** ] p-value <0.0001. Data depict single experiments that are representative of at least 2 independent studies for macrophage infections and 3 independent studies for *in vitro* experiments.

## The *bd* oxidase is necessary for optimal respiration under low pH conditions

The fitness defect of CydA-deficient bacteria in acidic pH suggested that under this condition, *bd* oxidase activity was increased, $bc_1/aa_3$ activity was decreased, or both. To investigate these possibilities, we used extracellular flux analysis (Agilent Seahorse XFe96) to measure oxygen consumption rate (OCR). We first optimized conditions to independently assess the activity of the two respiratory complexes. Treatment of WT H37Rv or the complemented Δ*cydA*::*cydABDC* strain with the cytochrome $bc_1/aa_3$-inhibitor, Q203 [5], led to a paradoxical increase in OCR (Fig 4A), which has been previously attributed to increased bd oxidase activity [7,10]. We confirmed this interpretation by finding that the Q203 treatment of the Δ*cydA* mutant virtually abolished OCR (Fig 4A). Thus, our Q203 treatment effectively inhibited $bc_1/aa_3$ and the OCR detected under these conditions solely reflected *bd* oxidase activity.

We next tested the effect of pH on each of the *bd*- and $bc_1/aa_3$-complexes. To interrogate the *bd* oxidase, we preadapted H37Rv to pH 7.4 or 4.5 and measured OCR during $bc_1/aa_3$ inhibition with Q203. We found that low pH accentuated *bd* oxidase-dependent OCR under this condition (Fig 4B). Treatment of H37Rv with the ATP synthase inhibitor, bedaquiline (BDQ) also lead to an expected increase in OCR [7], and was transiently enhanced by low pH (Fig 4B). To assess the effect of pH on the $bc_1/aa_3$ complex, we compared the OCR of H37Rv and Δ*cydA* strains. Preadaptation to pH 7.4 or 4.5 produced a steady-state OCR that was consistent over 25 minutes of monitoring. pH had little effect on the OCR of H37Rv or the Δ*cydA*::*cydABDC* strains, in which both respiratory complexes are functioning. In contrast, OCR of the Δ*cydA* mutant that exclusively depends on the $bc_1/aa_3$ system was markedly inhibited at pH 4.5 (Fig 4C). Thus, the *bd* oxidase is functional, and even operates at increased levels, at acidic pH; whereas the $bc_1/aa_3$ is inhibited under these conditions.

## Q203 is bactericidal at low pH

Given this pH-dependent decrease in $bc_1/aa_3$ activity, we hypothesized that acid stress would also increase the sensitivity of this complex to chemical inhibition. Indeed, reducing the pH from 7.4 to 5.5 enhanced the potency of Q203, lowering the $IC_{50}$ by almost 20-fold (Fig 5A). While Q203 has been found to be bacteriostatic at neutral pH *in vitro[29]*, we determined if the increased potency Q203 at low pH also produced bactericidal activity. H37Rv was grown at pH 7, pH 6.2, or pH 5.7 in the presence or absence of Q203 at 30x the $MIC_{50}$ for 6 days. Mtb viability was assessed by CFU at day 0, day 3, and day 6. At pH 7 and pH 6.2 Q203 showed the expected bacteriostatic effect, and no decrease in CFU was apparent in treated cultures (Fig 5B). However, at pH 5.7 we observed robust bactericidal activity for Q203, resulting in a 10-fold decrease in CFU by day 3 of treatment, and more than a 1,000-fold decrease by day 6 (Fig 5B). Thus, acidic pH has a dramatic effect on Q203 activity, both lowering its MIC and promoting cell death.

## Discussion

The flexibility of bacterial respiratory chains facilitates adaptation to changing environments, and in many situations the *bd* oxidase becomes critical under conditions where the $bc_1/aa_3$ complex is inhibited. In pathogens such as *E.coli*, *Listeria monocytogenes*, and *Salmonella typhimuirium*, the requirement for the cytochrome *bd* oxidase in bacterial virulence has been

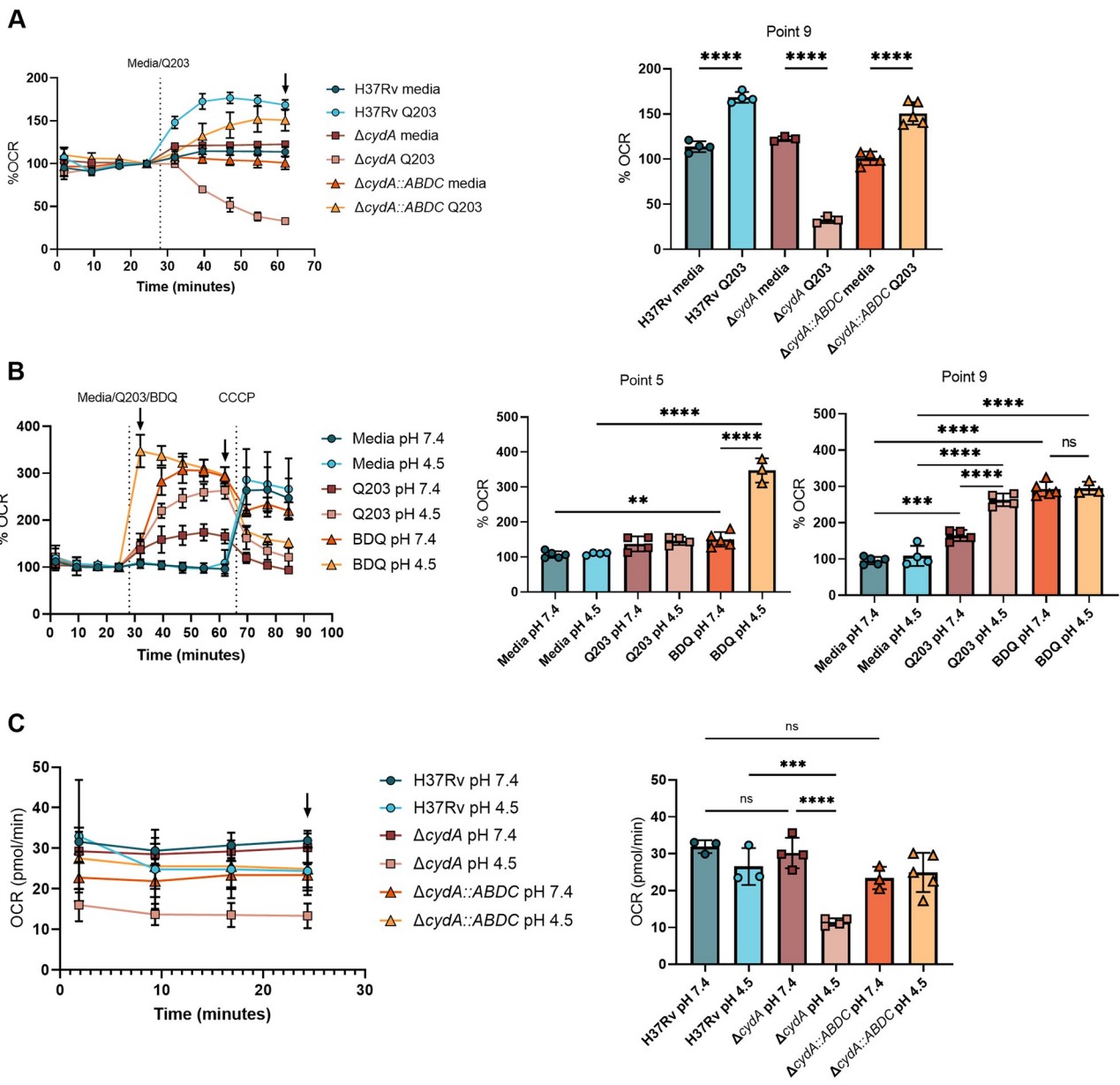

**Fig 4. The *bd* oxidase is necessary for optimal respiration under low pH conditions.** Oxygen consumption rate (OCR) was measured using the extracellular flux analyser. A) H37Rv, Δ*cydA* and Δ*cydA*::*ABDC* strains were exposed to media at pH 7.4 for ~30 minutes and then treated with Q203 (10nm) at the indicated time. Data are normalized to point 4. Bar graph of point 9 (black arrow). B) H37Rv was exposed to media at pH 7.4 and pH 4.5 for ~30 minutes and then treated with media, Q203 (300x $MIC_{50}$ 900nM), or BDQ (300x $MIC_{50}$ 16.2uM) followed by the uncoupler, CCCP, at the indicated times. Data are normalized to point 4. Bar graphs are plotted from point 5 and point 9 (black arrows in A) after the addition of BDQ/Q203 and before the addition of CCCP, respectively. C) OCR measurements (pmol $O_2$/min) of H37Rv, Δ*cydA* and Δ*cydA*::*ABDC* strains after ~30 minutes pre-exposure to media at pH 7.4 or pH 4.5. Bar graphs are plotted from point 4 (black arrow). Analysis was performed using a one-way ANOVA with Tukey post-test. * p-value < 0.05, ** p-value < 0.01, **** p-value <0.0001. Data depict single experiments that are representative of at least 2 independent studies.

attributed to its role in resisting hypoxia and nitrosative and oxidative stress [26,30–32]. While previous studies in *M. marinum* and *M. smegmatis* found that the mycobacterial *bd* oxidase can also confer resistance to hypoxia and peroxide, the specific roles played by the cytochrome *bd* and *bc₁*/*aa₃* oxidases of *Mtb* during infection has been less clear. Our work indicates that

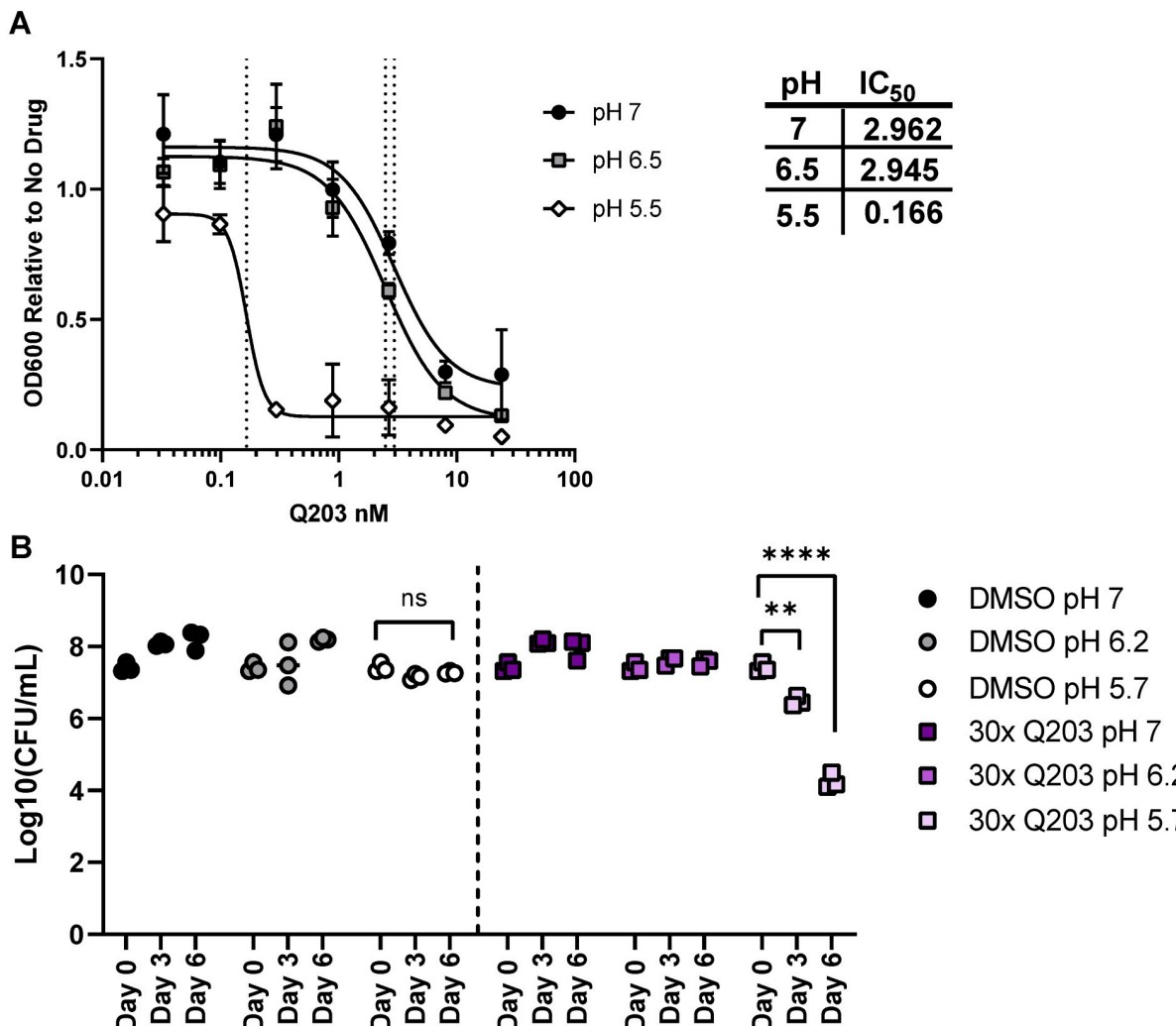

**Fig 5. Low pH increases the potency of Q203.** A) IC50 curves of H37Rv treated with Q203 at pH 7.0, pH 6.5 and pH 5.5 for 6 days in agitated culture. IC50 values for Q203 (nM) at each pH shown in the table. OD600 values are represented as relative to the no drug conditions because H37Rv growth is attenuated at pH 5.5. B) Day 0, 3, 6 CFU of H37Rv grown in agitated culture over 6 days at pH 7.0, 6.2, and 5.7 in the presence of 30x MIC Q203 (90 nM) or vehicle control (DMSO). Day 0 CFU timepoint is shared across all groups from DMSO pH 7 condition. Analysis was performed using a one-way ANOVA with Tukey post-test. ** p-value < 0.01, **** p-value <0.0001. Data depict single experiments that are representative of at least 3 independent studies.

the flexibility of the *Mtb* respiratory chain facilitates adaptation to changes in the immune response. As outlined below, our findings specifically suggest that the cytochrome *bd* oxidase provides resistance to IFNγ-mediated immunity by facilitating respiration under the acidic conditions encountered in the phagosomes of IFNγ-stimulated macrophages.

In both *ex vivo* macrophage cultures and intact animals, we found that the *bd* oxidase was required to resist IFNγ-dependent immunity. These data are consistent with those of *Shi* et al., who showed that *Mtb* cytochrome *bd* oxidase mutants were specifically attenuated in C57BL6 mice only after 50 days of infection. [17]. Conversely, other studies have concluded that the cytochrome *bd* oxidase is dispensable for growth in C57BL/6J and BALB/C mice [2,4,33]. Dhar & McKinney found that *cydC* is dispensable for growth in C57BL/6 mice but *cydC* mutants are attenuated during INH treatment in an IFNγ-dependent manner [33]. We suspect that these differing conclusions were caused by variations in the infection models. Specifically,

our study used a competitive infection, which ensures that both wild type and mutant bacteria are exposed to identical immune pressures and captures even transient differences in fitness. As a result, competitive studies, such as ours, are a particularly sensitive approach to detect differences in fitness. While it is possible that another animal model or an extended infection period would discover a more pronounced growth defect, even a small or transient difference in bacterial fitness reflects an evolutionary advantage that could be responsible for maintaining the *cydABDC* operon in the Mtb genome.

While IFNγ stimulation induces wide-spread transcriptional changes and antimycobacterial functions in macrophages, our data suggest that the *bd* oxidase requirement is related to phagosomal pH. NOS2 and Phox are strongly induced by IFNγ, and the requirement for the *bd* oxidase in other bacterial pathogens has been related to the resulting NO and ROS [24]. As a result, it was somewhat surprising that the sensitivity of Δ*cydA* mutant bacteria to IFNγ treatment was independent of these mediators. Instead, multiple lines of evidence indicated that this mutant was sensitive to the low pH encountered in the phagosome of IFNγ-stimulated macrophages. Firstly, we found that that *bd* oxidase-deficient bacteria grew poorly at low pH. These findings that are consistent with previous transposon mutant screening data suggesting that the *cydABDC* operon was required for optimal growth at pH 4.5 [16]. These *in vitro* growth defects were related to the intracellular growth environment by demonstrating that inhibition of vacuolar maturation and acidification with BAF abrogated the relative fitness difference between Δ*cydA* and wild type *Mtb* in IFNγ-stimulated macrophages. While these data are consistent with a primary role for the *bd* oxidase in adaptation to low pH conditions, we note that inhibition of phagosome acidification can have pleiotropic effects on processes such as phagosome-lysosome fusion and autophagy. Thus, while it remains possible that additional stresses play a role, our observations are consistent with a model in which the *bd* oxidase promotes resistance to the adaptive immune response by promoting respiration in the low pH environment of the IFNγ-stimulated macrophage.

While the requirement for *bd* oxidase activity at low pH can be attributed to the reduced activity we detected for the $bc_1/aa_3$ complex under these conditions, the mechanism by which low pH inhibits the activity of the cytochrome $bc_1/aa_3$ is unclear. The cytochrome $bc_1/aa_3$ super complex is tightly coupled to the transport of protons [34]. For every $O_2$ molecule reduced by the super complex, 4 protons are pumped into the periplasm and contribute to the proton motive force (PMF) [34]. While the cytochrome *bd* oxidase also contributes protons to PMF, it only contributes half of the protons for every molecular oxygen reduced as the super complex [8,35]. It is possible that the tight coupling between proton pumping and electron transfer for the cytochrome $bc_1/aa_3$ complex results in its inhibition when extracellular proton concentrations are high. However, acid stress induces a wide variety of transcriptional and physiological responses in *Mtb* and it is also possible that pH has additional indirect effects on the cytochrome $bc_1/aa_3$ complex [36–38].

The success of bedaquiline, a mycobacterial ATPase inhibitor, has made respiration an attractive target for new therapeutics. Multiple small molecule inhibitor screens have identified drugs that target the QcrB component of the proton-pumping cytochrome $bc_1/aa_3$ [3,5,39,40], most notably is Q203 (Telacebec) which is currently in clinical trials [5,6]. However, the flexibility of the mycobacterial respiratory chain has raised concerns about the potential efficacy of this drug [2,4,29,41]. One strategy to enhance the efficacy of respiratory inhibition is to simultaneously target both the $bc_1/aa_3$ and *bd* oxidase complexes, which produces a bactericidal effect [4,7,10]. Our data suggest that immunity is another important factor that determines the relative importance of terminal oxidases and the ultimate efficacy of these agents. The concept is similar to the previously described synergy between IFNγ-induced tryptophan depletion and the efficacy of *Mtb* tryptophan synthesis inhibitors [16]. These examples highlight the

importance of understanding the interactions between bacterial physiology and immunity for evaluating and optimizing new therapies.

## Experimental methods

### Ethics statement

Animal work was approved by University of Massachusetts Medical School IACUC (protocol number 202000009). All protocols conform to the USDA Animal Welfare Act, institutional policies on the care and humane treatment of animals, and other applicable laws and regulations.

### Bacterial growth and strain generation

*Mycobacterium tuberculosis* strains were cultured at 37˚C in complete Middlebrook 7H9 medium containing oleic acid-albumin-dextrose-catalase (OADC, Becton, Dickinson), 0.2% glycerol, and 0.05% Tween80 or 0.02% Tyloxapol. Hygromycin, kanamycin, and zeocin were add as necessary at 50 ug/mL, 25 ug/mL, and 25 ug/mL, respectively. All *Mtb* mutant strains were derived from the wildtype H37Rv. c*ydA* and c*ydABDC* operon were deleted by allelic exchange as described previously [22]. The gene deletions were confirmed by PCR verification and sequencing of the 5' and 3' recombinant junctions and the absence of an internal fragment within the deleted region. An L5attP-zeoR-CydABDC-operon complementing plasmid was assembled by Gateway reaction (Invitrogen) and transformed into the Δ*cydA*::*Hyg* mutant to generate the Δ*cydA*::*ABDC*-complementing strain. The Live/Dead reporter strains were generated by transforming *Mtb* with the replicating Live/Dead plasmid that contains a constitutively expressed GFP and a tetracycline-inducible TagRFP fluorescent protein.

### Mice

C57BL/6, *Cybb*$^{-/-}$, *Nos2*$^{-/-}$, *Ifngr1*$^{-/-}$ and *Rag2*$^{-/-}$ were purchased from the Jackson Laboratory. Housing and experimentation were in accordance with the guidelines set forth by the Department of Animal Medicine of University of Massachusetts Medical School and Institutional Animal Care and Use Committee. Animals used for experimentation were between 6 and 8 weeks old.

### Mouse infections

Prior to infection, *Mtb* strains were resuspended and sonicated in PBS containing 0.05% Tween80. Δ*cydA* mutant fitness *in vivo* was determined by inoculating mice with a ~1:1 mixture of Δ*cydA* (hygromycin resistant) and H37Rv (harboring pJEB402 chromosomally integrated plasmid encoding kanamycin resistance) strains via the respiratory route using an aerosol generation device (Glas-Col). At the indicated time points, mice were sacrificed and CFU numbers in lung homogenate were determined by plating on 7H10 agar supplemented with OADC containing Kanamycin (25 ug/mL) or Hygromycin (50 ug/mL).

### Macrophage infection

Bone marrow derived macrophages (BMDMs) were isolated from C57BL/6, *Cybb*$^{-/-}$ or *Nos2*$^{-/-}$ mice by culturing bone marrow cells in DMEM supplemented with 20% conditioned medium from L929, 10% FBS, 2 mM L-glutamine and 1 mM sodium pyruvate for 7 days. BMDMs were seeded and left unstimulated or stimulated with IFN-γ (25ng/mL, PeproTech) overnight and then infected with *Mtb* at an MOI of 5. After 4 h incubation, macrophages were washed twice with PBS to remove extracellular bacteria and incubated in fresh complete

medium with or without IFNγ. In some conditions, bafilomycin A (100ng/mL, Sigma) or Q203 (at specified concentrations) was added. Cells were lysed with 1% Saponin/PBS (Sigma) at 120 h after infection and then plated on 7H10-OADC plates in serial dilutions. CFUs were counted after 3 weeks of incubation at 37˚C.

## Flow cytometry

For flow cytometry, BMDMs pretreated with or without IFN-γ were infected with Live/Dead reporter *Mtb* strains. At day 3 post-infection, tetracycline (500 ng/ml) was added to medium. Macrophages were harvested after 24 hours tetracycline addition and fixed with 1% PFA for 45 minutes, then run on an LSR II flow cytometer.

## Acid sensitivity assays

*Mtb* strains in log-phase were wash twice with PBST (PBS + 0.05% Tween 80) and once with respective pH-adjusted media. For the 96-well plate assays, wells with 7H9-Tyloxapol-7.4, 7H9-Ty-6.0, 7H9-Ty-5.5, 7H9-Ty-5.0 and 7H9-Ty-4.5 were inoculated to a starting optical density at 600 nm ($OD_{600}$) of 0.01. Inkwells with 10mL of 7H9-Tyloxapol-7.4, 7H9-Ty-6.2, and 7H9-Ty-5.7 were inoculated to a starting optical density at 600 nm ($OD_{600}$) of 0.1. Citrate phosphate buffer or 2N NaOH were added to 7H9 medium containing oleic acid-albumin-dextrose-catalase (OADC, Becton, Dickinson), 0.2% glycerol, and 0.02% Tyloxapol until the desired pH was achieved. After a week in culture, the pH of the media was measured to ensure it was properly buffered. The pH 7 media was between pH 6.75–6.91, pH 6.2 media was between 6.23–6.30, and pH 5.7 media was between 5.60–5.70. There was no significant difference in media pH between H37Rv and Δ*cydA* cultures. In specified conditions, DMSO-reconstituted Q203 (gifted from Professor Barry Clifton, MedChemExpress Cat. No:HY-101040) was added to cultures at the indicated concentrations. The Syngery HXT microplate reader was used to measure daily $OD_{600}$ of 100uL aliquots in a 96 well plate.

## In vitro growth CFU assays

During *in vitro* assays, samples of each 10mL were collected on day 0, 3 (only in the Q203 assay), and 6. Samples were frozen in 15% glycerol. At the time of plating, samples were thawed and washed twice with PBST. Serial dilutions were done in PBST and plated on Middlebrook 7H10 agar supplemented with OADC (Fisher Scientific Cat. No. B12351) and glycerol. Plates for Δ*cydA* and Δ*cydA*::*ABDC* contained hygromycin (50ug/mL) or hygromycin (50ug/mL) + zeocin (25ug/mL), respectively. Plates were incubated at 37˚C for 3 weeks before counting CFU.

## Extracellular flux analysis

The OCR of Mtb bacilli adhered to the bottom of an XF cell culture microplate (Cell-Tak coated) (Seahorse Biosciences), at 2x106 bacilli per well, were measured using a XF96 Extracellular Flux Analyser (Seahorse Biosciences)[7]. All XF assays were carried out in unbuffered 7H9 media (pH 7.4 or pH 4.5 for acidic conditions) without a carbon source. Basal OCR was measured for ~ 25 min before the addition of compounds through the drug ports of the sensor cartridge. After media or Q203 addition (300x $MIC_{50}$ 0.9uM) or BDQ (300x $MIC_{50}$ 16.2uM), OCR was measured for ~ 40 min, followed by the addition of the uncoupler carbonyl cyanide m-chlorophenyl hydrazone (CCCP) (2 μM) and the OCR measured for a further ~20 min. All OCR Figures indicate the approximate point of each addition as dotted lines. OCR data points are representative of the average OCR during 4 min of continuous measurement in the

transient microchamber, with the error being calculated from the OCR measurements taken from at least three replicate wells by the Wave Desktop 2.2 software (Seahorse Biosciences). The microchamber is automatically re-equilibrated between measurements through the up and down mixing of the probes in the wells of the XF cell culture microplate.

## MIC assay

Log-phase H37Rv was washed twice with PBS + 0.02% tyloxapol and used to inoculate 10mL cultures of 7H9-Ty-7.0, 7H9-Ty-6.5, and 7H9-Ty-5.5 to an $OD_{600}$ of 0.02. As stated before, media pH was achieved by the addition of citrate-phosphate buffer or 2N NaOH. To determine the MIC of Q203 (Cat. No. HY-101040, MedChemExpress), 3-fold serial dilutions from 24nM to 0.3 nM were performed at pH 7.0, 6.5, and 5.5 with a vehicle (DMSO) control. Cultures were incubated at 37°C in inkwells with shaking. The Syngery HXT microplate reader was used to measure daily OD600 of 100uL aliquots in a 96 well plate. The MIC values were calculated on day 6 of growth using nonlinear regression analysis.

## Statistical analyses

Statistical tests and the number of replicate experiments performed are noted in each figure legend. For ANOVA analyses, the specific post-test used in each figure panel, was chosen based on the experimental design. Dunnet's post-test was used for comparing multiple experimental conditions to a single control sample. Sidak's post-test was used for comparing multiple pre-defined comparisons. Tukey's post-test was used if all samples were compared to each other.

## Author Contributions

**Conceptualization:** Yi Cai, Eleni Jaecklein, Jared S. Mackenzie, Andrew J. Olive, Adrie J. C. Steyn, Christopher M. Sassetti.

**Data curation:** Yi Cai, Eleni Jaecklein, Jared S. Mackenzie.

**Formal analysis:** Eleni Jaecklein, Jared S. Mackenzie, Adrie J. C. Steyn, Christopher M. Sassetti.

**Funding acquisition:** Andrew J. Olive, Christopher M. Sassetti.

**Investigation:** Yi Cai, Eleni Jaecklein, Jared S. Mackenzie.

**Methodology:** Yi Cai, Eleni Jaecklein, Jared S. Mackenzie, Kadamba Papavinasasundaram, Adrie J. C. Steyn, Christopher M. Sassetti.

**Project administration:** Christopher M. Sassetti.

**Resources:** Yi Cai, Jared S. Mackenzie, Kadamba Papavinasasundaram, Adrie J. C. Steyn.

**Supervision:** Xinchun Chen, Adrie J. C. Steyn, Christopher M. Sassetti.

**Validation:** Eleni Jaecklein.

**Visualization:** Yi Cai, Eleni Jaecklein.

**Writing – original draft:** Eleni Jaecklein, Christopher M. Sassetti.

**Writing – review & editing:** Eleni Jaecklein, Jared S. Mackenzie, Kadamba Papavinasasundaram, Adrie J. C. Steyn, Christopher M. Sassetti.

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
