## [Decision Letter · Decision Letter 0]

9 Sep 2020

Dear Dr. Sassetti,

Thank you very much for submitting your manuscript "Host immunity increases Mycobacterium tuberculosis reliance on cytochrome bd oxidase" for consideration at PLOS Pathogens. As with all papers reviewed by the journal, your manuscript was reviewed by members of the editorial board and by several independent reviewers. In light of the reviews (below this email), we would like to invite the resubmission of a significantly-revised version that takes into account the reviewers' comments.

The reviewers raise some comments that can easily be addressed (defining 100% OCR for the strains, etc). However, some concerns require additional work including defining actual CFU for critical experiments (CFU analysis for strains grown at low pH in presence/absence of Q203), analysis of the survival of the cyd mutant at low pH in the presence/absence of Q203 and preferably including the complemented strain in the flux analysis.

We cannot make any decision about publication until we have seen the revised manuscript and your response to the reviewers' comments. Your revised manuscript is also likely to be sent to reviewers for further evaluation.

Sincerely,

Helena Ingrid Boshoff

Associate Editor

PLOS Pathogens

Michael Wessels

Section Editor

PLOS Pathogens

Kasturi Haldar

Editor-in-Chief

PLOS Pathogens

orcid.org/0000-0001-5065-158X

Michael Malim

Editor-in-Chief

PLOS Pathogens

orcid.org/0000-0002-7699-2064

The reviewers raise some comments that can easily be addressed (defining 100% OCR for the strains, etc). However, some concerns require additional work including defining actual CFU for critical experiments (CFU analysis for strains grown at low pH in presence/absence of Q203), analysis of the survival of the cyd mutant at low pH in the presence/absence of Q203 and preferably including the complemented strain in the flux analysis.

Reviewer's Responses to Questions

**Part I - Summary**

Reviewer #1: Elucidating the role of the alternate branch of the aerobic electron transport chain in mycobacteria terminating in cytochrome bd oxidase has been a topic of considerable interest particularly in light of the discovery of QcrB as a vulnerable TB drug target and subsequent clinical development of Q203 as a new drug candidate. While there is abundant evidence of bd oxidase upregulation in mycobacteria in response to disruption of the main cytochrome bc1-aa3-terminating branch of the ETC, the specific cellular function/s served by the bd oxidase in the physiology of Mtb has remained elusive. This problem has been further compounded by discrepant findings on the impact of inactivating mutations in the cydABDC gene cluster on growth and persistence of Mtb during mouse infection with one earlier study revealing a persistence defect in late-stage infection, but others showing no discernible effect.

In this manuscript, the authors re-visit this question and demonstrate a role for the bd oxidase in resisting the host adaptive immune response via an atypical IFNg-dependent mechanism which does not rely on reactive nitrogen or oxygen radicals to inhibit the bd oxidase, but is manifest instead at low pH. In a significant advance, they establish a role for the bd oxidase in respiration under acidic conditions and accordingly show potentiation of anti-TB activity of Q203 at low pH. This is a clearly written manuscript that describes some important results of general interest.

Reviewer #2: The cytochrome bd oxidase is a bacteria-specific terminal oxidase involved in microaerophilic respiration and resistance to oxidative stress in several bacteria, including Mycobacterium tuberculosis. In this article, the author reports on the involvement of the cytochrome bd oxidase in resistance to host immunity in macrophages and in a mouse model of tuberculosis infection. Using an H37Rv �cydC strain, a series of experiments was conducted to demonstrate the requirement of the bd oxidase for optimal growth at acidic pH, for viability in IFN-g activated macrophages (in a NOS2- and Cybb-independent manner) and for respiration at low pH. The article has some merits but is to some extent difficult to read and evaluate given the lack of experimental details, heavy reliance on data normalization and -on few occasions- issues with data analysis. The rational for using several statistical methods throughout the study should be explained.

Reviewer #3: The paper by Sassetti and colleagues reports the role of cytochrome bd (cyd) of M. tuberculosis in resisting the adaptive immune response. The molecular basis for this adaptation was the role of cytochrome bd in protecting M. tuberculosis from the acidic pH of the IFN�-activated macrophages. The authors demonstrate that oxygen consumption by the cyd mutant is decreased at low pH because the bcc-aa3 proton-pumping complex is inhibited by an unknown mechanism. The manuscript is important because it again validates the respiratory oxidases as essential targets in TB drug development.

The paper is well written and the data presented in an easy to follow structure. However, the OCR work is a little disappointing as crucial pieces of information are missing that this referee requires to make a suitable recommendation.

**Part II – Major Issues: Key Experiments Required for Acceptance**

Reviewer #1: Line 185-187 and Fig. 4C and D. These refer to a ΔcydAB mutant, however this strain is not described in the Experimental Methods (line 264). A complemented derivative of the cydAB (or cydA?) mutant used in the flux analysis experiments shown in Fig. 4C and 4D must be included to confirm that the effect observed is attributable specifically to abrogation of bd oxidase function.

Reviewer #2: Figure 1

-Line 120/121: is the statement that addition of IFN-g significantly reduced the number of cells harboring live H37Rv supported by statistics (Fig 1b)?

- Mycobacterial viability in macrophages was monitored using a GFP/RFP reporter system, or CFU determination. FACS was used to determine the number of macrophages containing live bacteria (GFP+/RFP+) or dead bacteria (GFP+). Is the approach able to detect a cell containing both live and dead bacteria ? if not, this limitation should be discussed.

- line 125. The statement that the viability recorded with the report genes system (Fig 1b) correlates with CFU determination (Fig 1c) is not totally accurate since some statistical differences observed with the reporter genes were not confirmed by CFU determination. Please comment.

- The use of the ANOVA with Sidak post-test to analyze panel b and c, and the Dunnett’s multiple comparisons test for c need to be justified. What was the requirement to use ANOVA and the need for the Sidak post-test. How many times were the experiments repeated. Statistic should be applied to compared other conditions in panel d (e.g. H37Rv unstimulated vs H37Rv IFN-g stimulated or DcydD vs DcydD:cydABDC – both IFN-g activated).

- The need to normalize the results in the panel c & d should also be explained, why not presenting the CFU numbers instead of normalized values?

If the results are presented “as percent survival relative to untreated”, what does the standard deviation reflect in the untreated control groups?

Figure 2

- Co-infection was an excellent idea to compare the fitness of the cydA-deficient strain in animals. Few experimental details should be added to support the findings: inoculum size, number of animals per group, how many times the experiments were repeated, statistical method used. The bacterial load early after infection (e.g. 24 hours) should be shown to ensure that the implementation of both strains was in a similar range. If those data are not available, the enumeration of the ratio cydA-deficient/H37Rv in the inoculum would address this concern.

- A limitation of the study is the lack of one late time point to enhance the confidence that the differences (which are at best 5-fold reduction) at day 30 are robust.

- Lanbo Shi (PNAS 2005) reported that a H37Rv DcydC multiplied well in NOS2-/- mice, whereas this study suggests an attenuation phenotype. What could explain the difference with this previous report ?

Figure 3

- My main concern on this part of the work is the use of several assay formats to monitor growth. The assay in Fig 3a-b was done in 96 well plates (I assume the medium was 7H9-OADC-tyloxapol). This assay format may induce early oxygen depletion that could explain the requirement of the cyt-bd for optimum growth. The growth does not seem to be logarithmic, suggesting that the conditions are sub-optimum for growth.

- Growth rate should be calculated to confirm (using statistics) that the cydA KO multiplies at a slower rate at acidic pH. The pH of the culture broth media should be monitor overtime since the use of a phosphate buffer may not be appropriate to maintain the pH at 4.5.

- To rule out the influence of oxygen, the growth kinetics should be repeated at a higher constant oxygen tension (for instance in roller bottles or under slow agitation) to confirm the results. Comparing the results from Fig 3a-b with Fig 3c is not possible since the complementation studied was performed in a different assay format, and at only one time point. Fig 3c should be repeated in the same format as in Fig 3a-b and growth should be recorded at regular time interval to derive a growth rate.

- The macrophage experiments look convincing but should be confirmed (maybe in wt or NOS2-/- macrophages only) by CFU determination. It is a bit of as stretch to claim that the growth rescue of the cydA KO strain is a direct consequence of the increase in the pH of the phagosome: Bafilomycin A1 has multiple effects on the phagosome maturation that could explain the rescue. Please discuss the limitation of Bafilomycin A1.

Figure 4

- I have some difficulties understanding the design of the respirometry experiments. What seem to have been measured and discussed is not the effect of the pH on OCR, but the adaptive response after treatment with Q203.

- Instead of the % OCR (that seems to indicate that the baseline respiratory rates were normalized), it would be more appropriate to present the data in absolute unit (pmol O2 consumed/min). In additional experiments, the bacteria should also be pre-incubated at pH 7.4 and 4.5 for 30 to 60 min before OCR recording (in absolute unit) to give a better reflection of the effect of low pH on mycobacterial respiration. Alternatively, OCR recording could be extended to few hours.

- The statement “At pH 4.5, Q203-treated cells displayed an even higher OCR than at neutral pH, potentially indicating further inhibition of the cytochrome bc1/aa3 complex and increased reliance on cytochrome bd” is not accurate. It is not the OCR that is higher at low pH, but the extend of the OCR deregulation induced by Q203, which is a phenomenon not fully understood. The use of bedaquiline as a control would be a good addition.

- The interpretation of the results obtained with the cydA-deficient strain (lines 186-188) is peculiar. The mutant is already in a medium at either pH of 7.4 or 4.5. Injecting few ul of the same medium at the same pH should not trigger any change in OCR. I do not really follow how this observation could indicate that the bc1-aa3 is preferentially reduced under these conditions (lines 189-190). Was this experiment repeated more than once?

- As presented and analyzed, the respiratory experiments do not seem to support the conclusion that Mtb reliance on the cyt-bd is higher a low pH.

- The observation that Q203 has a lower MIC at pH 5.5 is interesting. For a direct comparison between the OCR and the MIC results, it would be nice to run both experiments at a similar pH.

- panel E. By default, it is a better practice to plot the raw OD600 values, what is the justification to normalize the values? Even if the bacteria multiply at a reduced rate at low pH, MIC50 can still be determined accurately. I would like to see few controls such as bedaquiline and rifampin to ensure that the shift in MIC is specific to Q203. To facilitate the experimental design, I would advise to use a microplate format (instead of the larger 10 ml volumes) and 2-fold drug dilution (8 points) to derive an accurate MIC50.

- There are few inconsistencies between the figure legends and the method section that need to be fixed. For instance, it is not clear if Q203 was used at 10 nM or at 900 nM in the respiratory experiments.

Reviewer #3: Specific comments for authors:

1. Fig 1A – did OD600 mirror cell viability (CFU) given the culture would have been experiencing hypoxia at day 12?

2. Did the authors perform an additional control for Fig. 1D by transforming CydABDC into the wild-type strain?

3. Fig. 3: please report the growth rate as a function of pH – calculated from logscale plots. The cydA mutant looks to be growing faster than WT? What is the actual difference here? In each of the pH values tested what was the final pH of the culture measured at 5 h? Does the pH increase or remain relatively constant? Same comment for Fig. 3C – what was growth rate?

4. Fig. 3: What does cell survival look like as a function of external pH? Do the cells also die at acidic pH and if so what is the mechanism of cell death – the obvious explanation is a defect in intracellular pH homeostasis. Can the authors comment on this?

5. Fig. 4: to make sense of this figure I need to know what 100% OCR is for the WT and CydAB mutant – are the rates the same? I need to see the data so I can make sense of the interpretation. Same comment applies to OCR at pH 7.4 and 4.5 –for wild-type and cyd what are the actual rates for 100% OCR?

6. Fig4E: why do the cells become more inhibited by Q203 at acidic pH if CydAB+ is still present? What does this data look like in the cyd mutant? Does Q203 (bacteriostatic) become bactericidal against the wild-type at acidic pH – this is an essential experiment.

7. Discussion page 9: why does low pH preferentially inhibit the bcc1-aa3 complex? You clearly show that the cyd mutant can still respire at acidic pH – need to know what the actual rate is in Fig. 4D to make any rational judgements.

8. Lines 78-80: cyd mutants of M. smegmatis are also hypersusceptible to bedaquiline and cydAB is induced by BDQ – cite Hards et al. J Antimicrob Chemother 2015; 70: 2028–2037 (reference 35 covers M. tb only)

9. Paper by the group of Rubin (ref 27) needs to be given a little more kudos for identifying the essential role for the cyd operon at acidic pH. Add to introduction.

**Part III – Minor Issues: Editorial and Data Presentation Modifications**

Reviewer #1: 1. Figure 1B and 1C, and text describing the experiments shown in this figure (lines 111-133). These figures show the statistical significance of comparisons between WT and ΔcydA strains. However, the more informative comparisons are between WT or mutant +/- IFNγ rather than between WT vs. mutant under a given condition (+/- IFNγ). Therefore, does either assay (Live/Dead or CFU) support the claim that the: “In wildtype BMDMs, addition of IFNg significantly reduced the number of cells harbouring live H37Rv and ΔcydA bacteria (Figure 1B)”, and if so, at what level of significance? Likewise, for the statement: “The CFU assay also showed that IFNγ treatment reduced the viability of H37Rv in a Nos2-dependent and Cybb-independent manner….”.

2. Line 82. Dhar & McKinney (PMC2901468) identified a transposon mutant in cydC in a screen for Mtb mutants that show accelerated clearance by INH in mice in a manner “dependent on environmental changes imposed by the IFN-γ-mediated immune response”. This paper is highly relevant to the work described in this study and must therefore be cited, and its implications for the work reported here discussed.

3. The authors attribute their ability to discern a persistence defect in IFNγ-competent mice to the fact that the experiment was done as a co-infection of WT and cydA mutant strains; even then, the persistence defect at day 30 is very modest (Fig. 2B). Is this due to the fact that in a competition assay, the cydA mutant has a competitive disadvantage against WT owing to an impaired ability to utilise (limited) oxygen during chronic infection? To broaden its appeal, the manuscript would also benefit from a fuller explanation of the implications of the findings for Mtb pathogenesis, including in the context of all the published literature. For example, what do the results reported here mean in terms of acidic (and hypoxic?) microenvironments in different mouse infection models? How does this differ in other animal models, in particular, NHPs? Can the authors speculate on why treatment of Mtb-infected marmosets with a bc1-aa3 inhibitor gave rise to cavitation, as reported by Beites et al.?

4. Line 79: Kana et al. (PMC95555) were the first to demonstrate induction of the cydABDC operon and a competitive growth disadvantage of a bd oxidase of M. smegmatis under hypoxia. This reference should be cited.

5. Line 187 - Reference is made to “green line and bar” but these are not shown.

6. Line 116: Replace "Figure 1B-C by "Figure 1B"

Reviewer #2: -

Reviewer #3: (No Response)

PLOS authors have the option to publish the peer review history of their article (what does this mean?). If published, this will include your full peer review and any attached files.

Reviewer #1: No

Reviewer #2: No

Reviewer #3: No
---

## [Editor Report · Decision Letter 1]

12 Jul 2021

Dear Dr. Sassetti,

We are pleased to inform you that your manuscript 'Host immunity increases Mycobacterium tuberculosis reliance on cytochrome bd oxidase' has been provisionally accepted for publication in PLOS Pathogens.

Best regards,

Helena Ingrid Boshoff

Associate Editor

PLOS Pathogens

Michael Wessels

Section Editor

PLOS Pathogens

Kasturi Haldar

Editor-in-Chief

PLOS Pathogens

orcid.org/0000-0001-5065-158X

Michael Malim

Editor-in-Chief

PLOS Pathogens

orcid.org/0000-0002-7699-2064

The authors have addressed all important concerns. This work is an important contribution to the field's understanding of the relative contribution of the different terminal oxidases to survival under different in vivo conditions.
---

## [Editor Report · Acceptance letter]

22 Jul 2021

Dear Dr. Sassetti,

We are delighted to inform you that your manuscript, "Host immunity increases *Mycobacterium tuberculosis* reliance on cytochrome *bd* oxidase," has been formally accepted for publication in PLOS Pathogens.

Best regards,

Kasturi Haldar

Editor-in-Chief

PLOS Pathogens

orcid.org/0000-0001-5065-158X

Michael Malim

Editor-in-Chief

PLOS Pathogens

orcid.org/0000-0002-7699-2064